# Virus and viroid diversity in hops, investigating the German hop virome

**Ali Pasha** [1], **Gritta Schrader** [2], **Heiko Ziebell** [1]*

1 Julius Kühn Institute (JKI) - Federal Research Centre for Cultivated Plants, Institute for Epidemiology and Pathogen Diagnostics, Braunschweig, Germany, 2 Julius Kühn Institute (JKI) - Federal Research Centre for Cultivated Plants, Institute for National and International Plant Health, Braunschweig, Germany

* heiko.ziebell@julius-kuehn.de

## Abstract

Germany is worldwide one of the largest hop (*Humulus lupulus* L.) producers, an essential crop for the brewing industry. However, infections caused by viruses and viroids can severely impact hop yield and quality. In 2019, citrus bark cracking viroid (CBCVd) – a highly aggressive pathogen in hop – was first reported in Germany, raising concerns about its spread and prompting a broader investigation of the German hop virome.To investigate the viro-diversity in German hops, we started with a pilot study in 2021 targeting three hopyards in the Hallertau region (Bavaria), where CBCVd was previously detected. This study was expanded in 2022 and 2023 to include other main hop growing regions of Tettnang (Baden-Wuerttemberg) and Elbe-Saale (Saxony, Saxony-Anhalt, Thuringia). Leaf samples were collected from hop as well as non-hop plants inside and outside the hopyard, pooled, and proceeded for double-stranded RNAs extraction. High-throughput sequencing (HTS) was used as a diagnostic tool, followed by RT-PCR confirmation. Our analysis identified four viruses infecting hops; hop latent virus (HpLV), hop mosaic virus (HpMV), apple mosaic virus (ApMV), arabis mosaic virus (ArMV) – and two viroids; hop latent viroid (HLVd) and CBCVd. HpLV, HpMV, and HLVd were consistently found across all targeted hopyards, while CBCVd was confined to the Hallertau region. ArMV was only detected in one hopyard at one sampling timepoint. ApMV was the only virus detected in both hop and non-hop plants. Additional analysis of hop pool datasets revealed the presence of other potential hop pathogens, i.e., fungi and bacteria. The results showed a low diversity of viruses and viroids infecting hops. However, this study provides a comprehensive overview on the major viruses and viroids in German hopyards. The results may serve as a useful resource for the development of disease management strategies in hop cultivation and highlight the valuable implementation of HTS in plant pathogen surveillance.

**Data availability statement:** The obtained datasets were archived in Sequence Read Archive (SRA) under Bioprojects: hop virome 2021: PRJNA1165201, hop virome 2022: PRJNA1167474, and hop virome 2023: PRJNA1167495. The obtained datasets were archived in Sequence Read Archive (SRA) under the Bioproject: PRJNA1254465. Variant analysis of the full genome sequence of CBCVd revealed eight CBCVd sequence variants, which were deposited in Genbank under accession numbers: PQ075924, PQ075925, PQ650572, PQ650573, PQ650574, PQ650575, PQ655430, and PQ655431.

**Funding:** The project was supported by funds of the Federal Ministry of Agriculture, Food and Regional Identity (BMLEH) based on a decision of the Parliament of the Federal Republic of Germany via the Federal Office for Agriculture and Food (BLE) under the innovation support programme (Grant number 2818714B19). The sequencing was funded by the Euphresco projects: VirusCurate 2019-E-312: Using High Throughput Sequencing to gain insights from virus collections and strengthening the infrastructure of Plant Virus Collections and 2020-A-347: Baseline studies on virus reservoirs.

**Competing interests:** The authors have declared that no competing interests exist.

## Introduction

Hop (*Humulus lupulus* L.) is a dioecious perennial climbing plant native to Europe, Asia, and North America [1]. It belongs to the family *Cannabaceae* and the genus *Humulus*, which contains three species: *Humulus japonicus*, *Humulus lupulus*, and *Humulus yunnanensis* [2,3]. Since the middle ages, hop has become an important additive for fermented brews due to their anti-microbial properties, and later it became the basic flavoring ingredient of beer [4].

This distinctive status was formally documented in 1516 by the Bavarian "*Reinheitsgebot*", which later became the current "Purity law" regulating beer production in Germany [4]. In 2024, Germany had almost 20,289 hectares of hop-growing area distributed in three main regions (Hallertau, Tettnang and Elbe-Saale), with a hop yield of 40,300 tons according to the Federal Ministry of Agriculture, Food and Regional Identity (BMELH) (https://www.bmel-statistik.de/landwirtschaft/bodennutzung-und-pflanzliche-erzeugung/hopfenanbau, accessed on May 14, 2025). Infections caused by viruses, viroids and various microorganisms, e.g., fungi and bacteria that reduce hop yield and quality have been reported in several studies [2,3,5]. Five viruses and two viroids affect hops worldwide and can cause significant losses in commercial hopyards: apple mosaic virus (ApMV), arabis mosaic virus (ArMV), and the three carlaviruses hop mosaic virus (HpMV), hop latent virus (HpLV), and American hop latent virus (AHLV), as well as hop latent viroid (HLVd) and hop stunt viroid (HSVd) [2]. In addition, several other viruses and viroids were reported with limited distribution or sporadic occurrence in hops [2]. Citrus bark cracking viroid (CBCVd) is an emerging threat to hops production; it was detected in severely stunted hop plants in Slovenia in 2007 [5]. CBCVd is an aggressive viroid on hops killing infected plants within a few years. Further outbreaks of CBCVd have been reported from Germany in 2019 [6] as well as Brazil in 2020 [7]. Hop plants have a vigorous annual growth resulting in high biomass. Automated pruning and harvesting of hops facilitates the easy transmission of viruses and viroids by mechanical means [2]. In addition, carlaviruses can be transmitted in a non-persistent manner by hop aphids (*Phorodon humuli*) [8], and ArMV can be transmitted by the nematode *Xiphinema diversicaudatum* [9] within and between hopyards. Traditional virus detection methods, such as ELISA or PCR, have limitations as they can only detect a few targets in one sample and require prior knowledge of the target pathogen [10]. The use of next-generation diagnostic tools such as high-throughput sequencing (HTS) enables the detection of multiple targets in a sample by generating massive amounts of sequencing data from native isolated DNAs/RNAs [10]. Over the past few years, the cost of HTS has dropped considerably, making it possible to use this technology for routine plant diagnostics [11,12]. HTS has significant potential in plant virus diagnostics, allowing comprehensive assessment of a plant's health status and distinguishing between virus variants that may contribute differently to disease etiology [13]. In addition, HTS has been used in viral metagenomics (viromics) studies to address important topics such as virus transmission among host reservoirs, the impact of agricultural activities on ecosystems and biodiversity, and the detection of new viruses in crops

and natural surroundings [14–17]. However, there are still limitations to using HTS as a diagnostic tool, including the need for high computing power and specialized personnel with bioinformatics expertise to analyze large sequencing datasets. To overcome the challenges of HTS data analysis, the e-Probe Diagnostic Nucleic-acid Analysis (EDNA) was developed and improved [18,19] and used in several studies to develop electronic probes (e-probes) for the *in silico* detection of a wide variety of known plant pathogens without the need for bioinformatics expertise [20–23]. In this study, an HTS-based approach has been applied to investigate the viro-diversity in German hops. A preliminary study to optimize sampling and processing strategies was started in 2021 targeting one region (Hallertau); the virome study was expanded in 2022 and 2023 to cover all main hop growing regions in Germany. To explore potential alternative reservoirs of viruses and viroids infecting hops, samples were also collected from non-hop plants within and surrounding the hopyards. This is the first virome study to investigate the viro-diversity of the German hops.

## Materials and methods

### Sampling

In 2021, samples were collected in the end of August from three hopyards in the Hallertau region in the southern of Germany (Fig 1A). The targeted hopyards had been infected with CBCVd as confirmed by the monitoring program of the Bavarian State Research Center for Agriculture, Bayerische Landesanstalt für Landwirtschaft (LfL) [24]. Plant leaf material was collected randomly from each hopyard, whether symptomatic or asymptomatic, as follows: Ten hop samples (consisting each of three leaves from one plant, ranging from the fully matured to the newly emerged leaves), ten non-hop samples from within the hopyards; including grasses and weeds growing among the hop rows, and ten non-hop samples from weeds and wild plants growing around the hopyard (≤ 50 meter around the hopyard).

In 2022 and 2023, samples were collected at the end of June – beginning of July from two hopyards in each hop growing region. The hop growing locations were: Hallertau region, Tettnang region, and Elbe-Saale (ranging from Halle to Dresden) as shown in Fig 1B.

In those two years, the samples from each hopyard included: 35 hop samples, 5 non-hop samples within the hopyard, and 10 non-hop samples outside the hopyard (Fig 1B). Samples were collected from hopyards cultivated with different hop varieties and applying different agricultural practices (Table 1); and for sampling a "W" pattern was applied as suggested by [25]. Detailed information on the plant samples collected is provided in S1–S3 Tables.

### Pooling of samples

Plant leaf material (200 mg) from each sample collected from a hopyard was pooled with other samples according to the pooling strategy (Fig 2 and Table 2). In addition, all hop samples collected each year from all hopyards were pooled, as well as all non-hop plant samples from inside and outside the hopyards, resulting in a total of six pools (2021) and nine pools (2022 and 2023) (Fig 2). Samples were ground under liquid nitrogen using a mortar and pestle and then collected in 50 ml Falcon tubes. The tubes were stored at −80°C for subsequent RNA extraction. The remaining sample tissues were stored at −20°C.

### Nucleic acid extraction, virus/viroid enrichment and high-throughput sequencing

Double-stranded RNA (dsRNA) was extracted from each sample pool using the Double-RNA Viral dsRNA Extraction Mini Kit (plant tissue) (iNtRON Biotechnology, USA). A total of 200 mg of ground plant material was used as the starting material according to the manufacturer's instructions. A different dsRNA enrichment protocol had to be used for the 2023 samples as the iNtRON kit was no longer commercially available. Total RNA was extracted from all sample pools. Extraction was performed using the Monarch® Total RNA Miniprep Kit (New England Biolabs, Ipswich, USA) according to the manufacturer's instructions. For enrichment of viral and viroidal RNAs, total RNAs were enzymatically treated using a mix of

## A    Sampling 2021

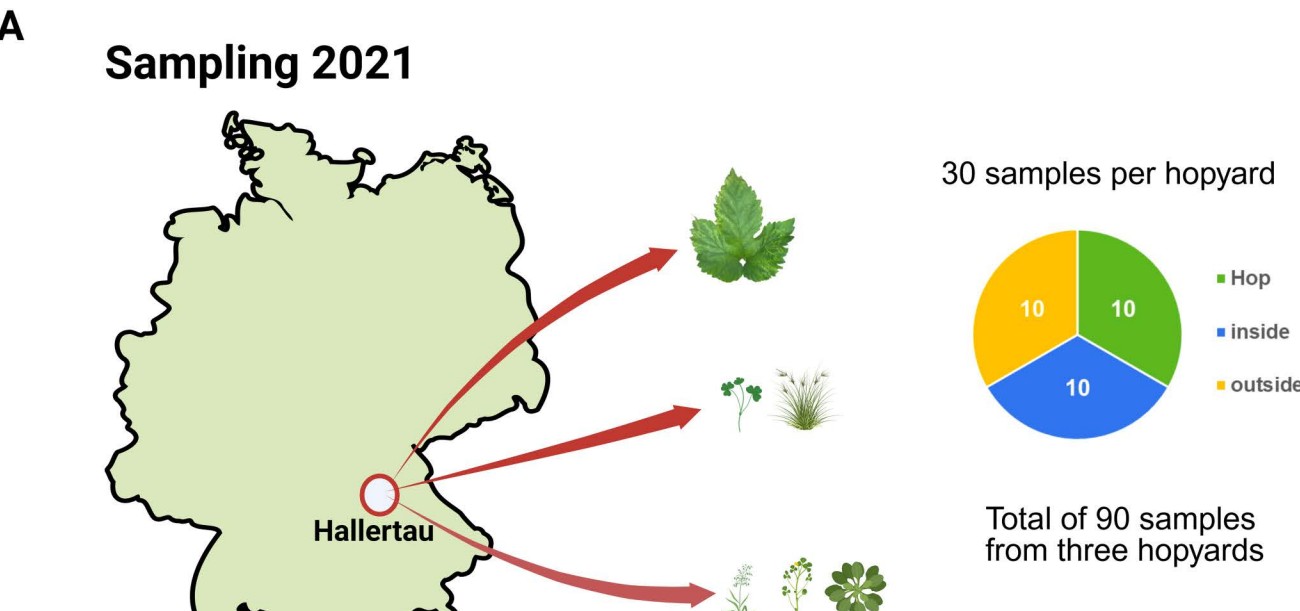

## B    Sampling 2022 and 2023

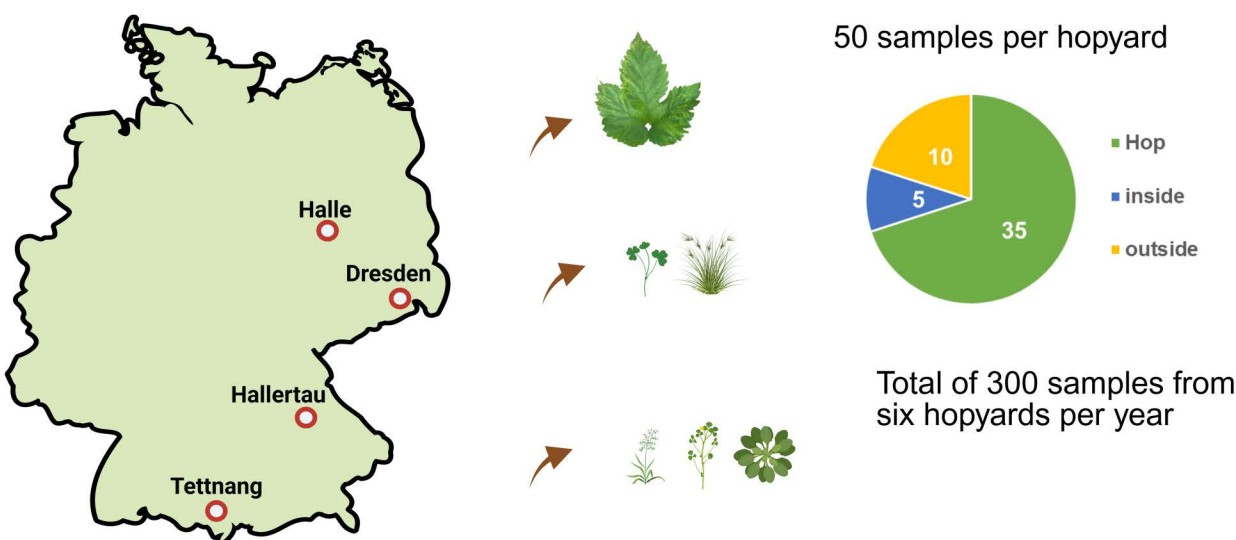

**Fig 1. Sampling locations in 2021 (A), 2022 and 2023 (B).** The type of sample and the number of samples of each type collected per yard. Created in BioRender. Pasha, A. (2025) https://BioRender.com/6o9sv6f.

Table 1. Detailed information on sampling locations. Samples were collected from hopyards 1 to 3 during the 2021 sampling, and from hopyards 4 to 9 in both 2022 and 2023 samplings.

| Sampling year | Hopyard | Hop variety | Farming type | Sampling location | | | | |
|---|---|---|---|---|---|---|---|---|
| | | | | Hallertau | | Tettnang | Elbe-Saale | |
| | | | | Geisenfeld | Rohrbach | Meckenbeuren | Burkau | Quellendorf |
| 2021 | Hopyard 1 | Perle | Conventional | X | | | | |
| | Hopyard 2 | Herkules | Conventional | X | | | | |
| | Hopyard 3 | Herkules | Conventional | X | | | | |
| 2022 and 2023 | Hopyard 4 | Herkules | Conventional | | X | | | |
| | Hopyard 5 | Perle | Conventional | X | | | | |
| | Hopyard 6 | Tettnanger | Conventional | | | X | | |
| | Hopyard 7 | Spalter, Tettnanger, Perle | Organic | | | X | | |
| | Hopyard 8 | Magnum | Conventional | | | | X | |
| | Hopyard 9 | Magnum | Conventional | | | | | X |

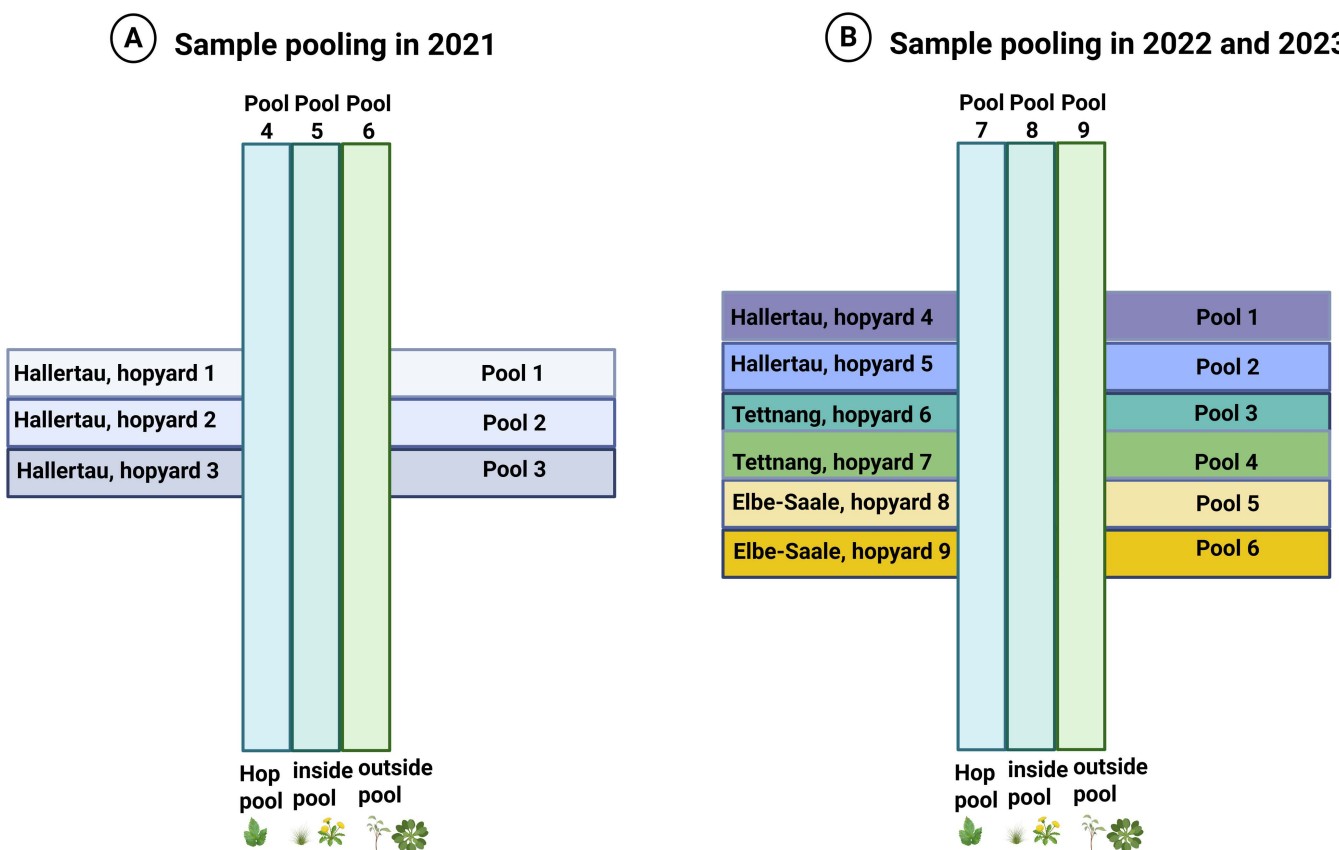

**Fig 2. Pooling strategy for all samples.** Collected samples from each yard were properly pooled in the hopyard pool. The hop samples collected in each sampling year were sampled in an additional pool called "hop pool", and the same was done for non-hop plants collected from inside and outside the hop yard, resulting in "non-hop inside pool" and "non-hop outside pool". Created in BioRender. Pasha, **A.** (2025) https://BioRender.com/ts4jf0m.

**Table 2. Sample pool.**

| Sampling year | Pool type | Amount of each sample (mg) | Number of pooled samples | Total amount of plant material (mg) |
|---|---|---|---|---|
| **2021** | Hopyard pool | 200 | 30 | 6000 |
| | Hop pool | 200 | 30 | 6000 |
| | Non-hop inside pool | 200 | 30 | 6000 |
| | Non-hop outside pool | 200 | 30 | 6000 |
| **2022 and 2023** | Hopyard pool | 200 | 50 | 10000 |
| | Hop pool | 200 | 210 | 42000 |
| | Non-hop inside pool | 200 | 30 | 6000 |
| | Non-hop outside pool | 200 | 60 | 12000 |

DNase I (Thermo Scientific) at a final concentration of 10 U/μL and RNase T1 (Thermo Scientific) at a final concentration of 10 U/μL for 30 minutes at 37°C. The undigested double-stranded viral/viroid RNAs (dsRNAs) were then purified using the RNA Mini Elute Kit (Qiagen, Hilden, Germany) [26]. The quantification of dsRNA was done by Qubit spectrofluorometry using the Qubit dsDNA high sensitivity kit (Qubit fluorometer, Invitrogen, Life Technologies). Samples were sequenced using the Illumina platform (NovaSeq 6000) to obtain 150 bp paired-end reads (INVIEW Virus, Eurofins Genomics, Constance, Germany). Ten million reads were ordered per sample. The obtained datasets were archived in the Sequence Read Archive (SRA) under Bioprojects: hop virome 2021: PRJNA1165201, hop virome 2022: PRJNA1167474, and hop virome 2023: PRJNA1167495. More details on bioproject datasets are provided in S4 Table.

For amplicon sequencing of fungal and bacterial sequences, DNA was extracted from hop pool samples using the DNeasy Plant Mini Kit (QIAGEN, Germany) following the manufacturer's instructions. DNA samples were sent to a commercial service provider for amplicon sequencing targeting the internal transcribed spacer 1 (ITS1) region to identify the fungal sequences, and 16S ribosomal RNA to identify the bacterial sequences. Samples were sequenced using the Illumina platform (MiSeq) to obtain 300 bp paired-end reads (Microbiome Profiling, Eurofins Genomics, Constance, Germany). The obtained datasets were archived in the Sequence Read Archive (SRA) under the Bioproject: PRJNA1254465.

## Bioinformatics analysis

The raw fastq datasets were imported into Geneious Prime® version 2024.0.4. Reads were paired and then quality trimmed (including removal of low quality reads and duplicated reads) using the BBNorm tool version 38.84 using default settings. *De novo* assembly for the normalized reads was performed using "Geneious" assembler: "medium sensitivity/fast" default settings. The obtained contigs were mapped against a local virus/viroid reference sequences (RefSeq Nucleotide) database downloaded in September 2024 from the National Center for Biotechnology Information (NCBI Virus) GenBank non-redundant nucleotide database. Contigs matching virus/viroid sequences were re-blasted with BLASTn and BLASTx using the National Center for Biotechnology Information (NCBI) GenBank non-redundant nucleotide and protein databases, respectively. Based on the BLAST results, virus or viroid GenBank sequences matching the consensus sequences with the highest similarity were used as reference genomes for re-mapping of quality-trimmed reads.

Mapping normalized reads to individual reference genomes was performed in Geneious Prime® version 2024.0.5. using Geneious RNA mapper (High sensitivity/Medium) with adjusting the settings: 'Trim before mapping "Do not trim". Results "save in sub-folder"'.

For confirmation of the identified virus and viroid consensus sequences in the original plant samples, (RT-)PCR primers were designed using a modified version of Primer3 (version 2.3.7) within Geneious Prime®.

Variant analysis of viroid sequences was performed using the "Find Variations/SNPs" tool in Geneious Prime version 2024.0.5. A minimum variant frequency threshold of 0.2 (20%) was applied; at least 20% of the reads had to support a

variant. To ensure reliability, a minimum coverage of 5 reads at the SNP position was also set, and the variants were manually inspected in the alignment viewer.

Pairwise nucleotide alignments were performed with MAFFT (Multiple Alignment using Fast Fourier Transform) version 7.490 within Geneious Prime®. Phylogenetic analysis was performed using the MrBayes version 3.2.6 [27].

The fastq datasets acquired by amplicon sequencing were analyzed by the sequencing service provider (Eurofins Genomics, Constance, Germany) using their microbiome analysis pipeline as follows: Ambiguous and chimeric reads were identified and removed based on the *de-novo* algorithm of UCHIME [28] as implemented in the VSEARCH package [29]. High quality reads were processed using minimum entropy decomposition (MED) [30]. To assign taxonomic information to each Operational Taxonomic Unit (OTU), DC-MEGABLAST alignments of cluster-representative sequences to the sequence database were performed. A most specific taxonomic assignment for each OTU was then transferred from the set of best-matching reference sequences (lowest common taxonomic unit of all best hits). A minimum sequence identity of 70% over at least 80% of the representative sequence was used as the minimum requirement for consideration of reference sequences. Further processing of OTUs and taxonomic assignments was performed using the QIIME software package (version 1.9.1). An analysis report was submitted for each sample containing the results (taxonomic composition of the sample).

### RT-PCR confirmation of viral sequences

To confirm that the identified virus/viroid sequences were indeed of viral origin, total RNA was re-extracted from all sample pools using the Monarch® Total RNA Miniprep Kit. Samples were subjected to RT-PCR using OneTaq One-Step RT-PCR, NEB kit following the manufacturer's instructions. The PCR conditions were as follows: reverse transcription 48°C for 15 min, initial denaturation 94°C for 1 min, denaturation 94°C for 15 sec, appropriate annealing temperature for the primer set in use (S5 Table) for 30 sec, extension at 68°C for 60 sec (40 cycles) followed by a final extension 68°C for 5 min, then hold at 4°C. The amplicons were purified using DNA clean-up and concentrator kit (Zymo Research) according to the manufacturer's instructions and sequenced using the same RT-PCR primers by Sanger sequencing (Microsynth, Germany). The primers used for the PCR assays from this study, as well as those from other studies, are listed in S5 Table.

## Results

### Sequencing data quality

A substantial higher amount of raw sequence reads was received by the sequencing provider, in some cases almost ten times more reads were produced. However, most of these "extra" reads were duplicated or of low quality and were not used for downstream bioinformatics analyses. The double-stranded RNA enrichment strategy resulted in RNA qualities that were interpreted by the sequence provider as insufficient for sequencing after library preparation so that the same library was sequenced multiple times introducing a high number of technical duplicate reads, and increasing the total number of raw reads artificially (S4 Table). Only reads after QC were used for *de novo* assembly and reference mapping producing sufficient read depth and coverage for virus and viroid detection (S4 Table, S6 Table).

### Identified viruses and viroids in hops across the sampling sites over three years

Four viruses and two viroids were identified in the hop pools over three years of sampling. These viruses belong to three virus families (*Betaflexiviridae*, *Bromoviridae*, and *Secoviridae*). The two identified viroids belong to *Pospiviroidae* family (Table 3 and 4).

HpLV, HpMV, and HLVd were consistently identified in all hopyard pools across all the sampling sites over three years (Table 3 and S6 Table). Based on our sampling strategy, we could assign the identification of ApMV to one hopyard pool in 2021, three hopyard pools in 2022 and four hopyard pools in 2023 (Table 3). ArMV was identified only in one hopyard pool in 2023. CBCVd was identified in three hopyard pools in 2021, in one hopyard pool in 2022 and in two hopyard pools in 2023 (Table 3).

**Table 3. Identified viruses and viroids in all sample pools over three years.**

| | Sampling year | 2021 | | | 2022 | | | | | | 2023 | | | | | |
|---|---|---|---|---|---|---|---|---|---|---|---|---|---|---|---|---|
| | Hopyard no. | 1 | 2 | 3 | 4 | 5 | 6 | 7 | 8 | 9 | 4 | 5 | 6 | 7 | 8 | 9 |
| **Hop pools** | Hop latent virus | ■ | ■ | ■ | ■ | ■ | ■ | ■ | ■ | ■ | ■ | ■ | ■ | ■ | ■ | ■ |
| | Hop mosaic virus | ■ | ■ | ■ | ■ | ■ | ■ | ■ | ■ | | ■ | ■ | ■ | ■ | ■ | |
| | Apple mosaic virus | | | ■ | | ■ | | ■ | | | ■ | | ■ | | | ■ |
| | Arabis mosaic virus | | | | | | ■ | | | | | | | | | |
| | Hop latent viroid | ■ | ■ | ■ | ■ | ■ | ■ | ■ | ■ | ■ | ■ | ■ | ■ | ■ | ■ | ■ |
| | Citrus bark cracking viroid | ■ | ■ | ■ | | | | | | | ■ | ■ | | | | |
| | | | | | | | | | | | | | | | | |
| **Non-hop inside** | Barley virus G | | | ■ | | | | | | | | | | | | |
| | Turnip ringspot virus | | ■ | ■ | | | | | | | | | | | | |
| | Raphanus sativus cryptic virus 1 | ■ | | | | | | | | | | | | | | |
| | Plant associated deltapartitivirus 1 | | | | ■ | | | | | | | ■ | | ■ | | ■ |
| | Turnip mosaic virus | | | | | | | | | | | | | | | ■ |
| | Turnip yellows virus | | | | | | | | | | | | | ■ | | |
| | Beet western yellows virus associated RNA | | | | | | | | | | | | | | | |
| | Plantago yellow mosaic virus | | | | | | | | | | | | ■ | ■ | | |
| | Turnip yellows virus associated RNA2 | | | | | | | | | | | | | ■ | | |
| | Raphanus sativus cryptic virus 4 | | | | | | | | | | | | | ■ | | |
| | Radish mosaic virus | | | | | | | | | | | | | ■ | | |
| | | | | | | | | | | | | | | | | |
| **Non-hop outside** | Turnip yellow mosaic virus | | ■ | | | | | | | | | | | | | |
| | Turnip ringspot virus | | ■ | ■ | | | | | | | | | | | | |
| | White clover mosaic virus | | | | | | ■ | | | | | | | ■ | | |
| | Plant associated deltapartitivirus 1 | | | | ■ | | | | | | | | | | | |
| | Apple mosaic virus | | | | | | | ■ | | | | | | | | |
| | Plantago yellow mosaic virus | | | | | ■ | | | | | | | | ■ | ■ | |
| | Peanut stunt virus | | | | | | | | | | | | | ■ | | |
| | Prune dwarf virus | | | | | | | | | | | | | | | ■ |
| | Senna severe yellow mosaic virus | | | | | | | | | | | | | | | ■ |
| | Mentha macluravirus 1 | | | | | | | | | | ■ | | | | | |

**Viruses and viroids identified in non-hop plants.** Nine viruses belonging to four virus families in addition to two virus associated RNAs were identified in the non-hop plant samples collected from inside the hopyard over three years (Table 3 and 5). The plant diversity of non-hop plants within the hopyards was low with 90 plant samples belonging to 11 plant families.

In contrast, the sample pools representing plants from outside the maintained hopyards had a wider variety of plant species, as a total of 150 plant samples belonging to 25 plant families were collected from outside the hopyards over three years (S1, S2, and S3 Tables). This plant diversity is also represented by a higher diversity of viruses that were identified from those plants: ten viruses belonging to six families (Table 6).

## Commonalities and specificities between the different sites/pools

The analysis of non-hop plant pools collected from inside and outside the hopyards over three years could only identify one virus in both hop and non-hop plants, ApMV. ApMV was consistently detected in hop pools over three years of

**Table 4. Identified viruses and viroids in hop pools over three years of sampling.**

| Year | Sample | Virus/Viroid | Family, genus | Ref. genome | Pairwise ID % (BLASTn) | Pairwise ID % (BLASTx) | Genbank accession |
|---|---|---|---|---|---|---|---|
| **2021** | **Hop pool** | Hop latent virus | *Betaflexiviridae, Carlavirus* | KP861891 | 98.12 | 98.68 | PV364036 |
| | | Hop mosaic virus | *Betaflexiviridae, Carlavirus* | EU527979 | 99.95 | 99.95 | PV364039 |
| | | Apple mosaic virus RNA 1 | *Bromoviridae; Ilarvirus* | NC_003464 | 99.75 | 99.81 | PV364044 |
| | | Apple mosaic virus RNA 2 | *Bromoviridae; Ilarvirus* | NC_003465 | 90.7 | 92.8 | PV364045 |
| | | Apple mosaic virus RNA 3 | *Bromoviridae; Ilarvirus* | NC_003480 | 95.91 | 96 | PV364048 |
| | | *Hop latent viroid | *Pospiviroidae, Cocadviroid* | MK795526 | 99.22, 98.83, 95.7 | | PV364051, PV364052, PV364053 |
| | | *Citrus bark cracking viroid | *Pospiviroidae, Cocadviroid* | KM211546 | 99.3, 99.65 | | PQ650572, PQ650573 |
| **2022** | **Hop pool** | Hop mosaic virus | *Betaflexiviridae, Carlavirus* | EU527979 | 99.95 | 99.95 | PV364040 |
| | | Hop latent virus | *Betaflexiviridae, Carlavirus* | KP861891 | 99.71 | 99.5 | PV364037 |
| | | Apple mosaic virus RNA 1 | *Bromoviridae; Ilarvirus* | NC_003464 | 98 | 99.04 | PV364042 |
| | | Apple mosaic virus RNA 2 | *Bromoviridae; Ilarvirus* | NC_003465 | 98.51 | 99.3 | PV364046 |
| | | Apple mosaic virus RNA 3 | *Bromoviridae; Ilarvirus* | NC_003480 | 97.73 | 97.88 | PV364049 |
| | | Arabis mosaic virus RNA2 | *Secoviridae, Nepovirus* | BK059335 | 85.96 | 90.38 | PV364058 |
| | | Hop latent viroid | *Pospiviroidae, Cocadviroid* | MK795526 | 99.61, 99.22 | | PV364054, PV364055 |
| | | Citrus bark cracking viroid | *Pospiviroidae, Cocadviroid* | KM211546 | 98.94, 98,94 | | PQ650574,PQ650575 |
| **2023** | **Hop pool** | Hop mosaic virus | *Betaflexiviridae, Carlavirus* | ON409684 | 99.21 | 99.79 | PV364041 |
| | | Hop latent virus | *Betaflexiviridae, Carlavirus* | KP861891 | 99.3 | 99.49 | PV364038 |
| | | Apple mosaic virus RNA 1 | *Bromoviridae; Ilarvirus* | NC_003464 | 97.09 | 98.15 | PV364043 |
| | | Apple mosaic virus RNA 2 | *Bromoviridae; Ilarvirus* | NC_003465 | 98.2 | 98.7 | PV364047 |
| | | Apple mosaic virus RNA 3 | *Bromoviridae; Ilarvirus* | NC_003480 | 97.73 | 98 | PV364050 |
| | | Hop latent viroid | *Pospiviroidae, Cocadviroid* | MK795539 | 99.61, 99.22 | | PV364056, PV364057 |
| | | Citrus bark cracking viroid | *Pospiviroidae, Cocadviroid* | KM211546 | 99.65, 98.94, 98.25, 98.59 | | PQ075924, PQ075925, PQ655430, PQ655431 |

*Viroid sequences identified in the hopyard and hop pools datasets were subjected to variant analysis. The identified variants were submitted to GenBank. The BLASTn results for the variants are separated by commas and are in the same order as the GenBank accessions in the right column.

**Table 5. Identified viruses in weeds collected from inside the hopyards "non-hop inside" over three years.**

| Year | Sample | Virus/Viroid | Family, genus | Ref. genome | Pairwise ID % (BLASTn) | Pairwise ID % (BLASTx) |
|---|---|---|---|---|---|---|
| **2021** | **Non-hop inside** | Barley virus G | *Solemoviridae, Polerovirus* | NC_029906 | 98.5 | 99.35 |
| | | Turnip ringspot virus RNA1 | *Secoviridae, Comovirinae* | GQ222381 | 94.9 | 95.13 |
| | | Turnip ringspot virus RNA 2 | *Secoviridae, Comovirinae* | FJ516746 | 95.8 | 96.2 |
| | | Raphanus sativus cryptic virus 1, dsRNA1 | *Partitiviridae, unclassified* | NC_008191 | 99.7 | 99.83 |
| | | Raphanus sativus cryptic virus 1, dsRNA2 | *Partitiviridae, unclassified* | NC_008190 | 95.4 | 97 |
| **2022** | **Non-hop inside** | Plant associated deltapartitivirus 1 RNA 1 | *Partitiviridae, Deltapartitivirus* | OL472010 | 99.5 | 99.79 |
| | | Plant associated deltapartitivirus 1 RNA 2a | *Partitiviridae, Deltapartitivirus* | OL472011 | 99.2 | 99.79 |
| | | Plant associated deltapartitivirus 1 RNA 2b | *Partitiviridae, Deltapartitivirus* | OL472012 | 99.4 | 99.98 |
| **2023** | **Non-hop inside** | Turnip mosaic virus | *Potyviridae, Potyvirus* | AB989659 | 98.5 | 98.9 |
| | | Turnip yellows virus | *Solemoviridae; Polerovirus* | OP797722 | 95.4 | 97.2 |
| | | Plant associated deltapartitivirus 1 RNA 1 | *Partitiviridae, Deltapartitivirus* | OL472010 | 99.7 | 100 |
| | | Plant associated deltapartitivirus 1 RNA 2a | *Partitiviridae, Deltapartitivirus* | OL472011 | 99 | 99.8 |
| | | Plant associated deltapartitivirus 1 RNA 2b | *Partitiviridae, Deltapartitivirus* | OL472012 | 99.5 | 100 |
| | | Beet western yellows virus associated RNA | *Riboviria;* virus-associated RNAs | ON603912 | 93 | 94.2 |
| | | Plantago yellow mosaic virus | *Riboviria,* unclassified | ON012585 | 99.5 | 99.86 |
| | | Turnip yellows virus associated RNA2 | *Solemoviridae; Polerovirus* | MN497827 | 95.4 | 96.3 |
| | | Raphanus sativus cryptic virus 4 RNA1 | *Partitiviridae, Durnavirales* | MF686921 | 97.6 | 98 |
| | | Raphanus sativus cryptic virus 4 RNA2 | *Partitiviridae, Durnavirales* | MF686922 | 99.6 | 99.7 |
| | | Radish mosaic virus, RNA1 | *Comovirinae; Comovirus* | NC_010709 | 99.1 | 99.4 |
| | | Radish mosaic virus, RNA2 | *Comovirinae; Comovirus* | NC_010710 | 99.2 | 99.34 |

**Table 6. Identified viruses in weeds and wild plants collected from outside the hopyards "non-hop outside" over three years.**

| Year | Sample | Virus/Viroid | Family, genus | Ref. genome | Pairwise ID (BLASTn) | Pairwise ID (BLASTx) |
|---|---|---|---|---|---|---|
| **2021** | **Non-hop-outside** | Turnip yellow mosaic virus | *Tymoviridae; Tymovirus* | NC_004063 | 92 | 93.7 |
| | | Turnip ringspot virus RNA1 | *Secoviridae; Comovirinae* | GQ222381 | 94.1 | 94.5 |
| | | Turnip ringspot virus RNA 2 | *Secoviridae; Comovirinae* | FJ516746 | 91 | 92.8 |
| **2022** | **Non-hop-outside** | White clover mosaic virus | *Alphaflexiviridae; Potexvirus* | OL472250 | 94 | 94.6 |
| | | Plant associated deltapartitivirus 1 RNA 1 | *Partitiviridae, Deltapartitivirus* | OL472010 | 99.2 | 99.7 |
| | | Apple mosaic virus RNA3 | *Bromoviridae; Ilarvirus* | NC_003480 | 94.7 | 96 |
| | | Plantago yellow mosaic virus | *Riboviria,* unclassified | OK523415 | 98.4 | 98.65 |
| **2023** | **Non-hop-outside** | Plantago yellow mosaic virus | *Riboviria,* unclassified | ON012585 | 96.2 | 97 |
| | | White clover mosaic virus | *Alphaflexiviridae, Potexvirus* | MN399743 | 93.8 | 94.3 |
| | | Peanut stunt virus RNA 1 | *Bromoviridae; Cucumovirus* | OK558739 | 98.8 | 99.2 |
| | | Peanut stunt virus RNA 2 | *Bromoviridae; Cucumovirus* | OR233184 | 98.3 | 98.8 |
| | | Peanut stunt virus RNA 3 | *Bromoviridae; Cucumovirus* | OR233185 | 98.5 | 98.76 |
| | | Prune dwarf virus RNA1 | *Bromoviridae; Ilarvirus* | MZ291947 | 98.2 | 98.45 |
| | | Prune dwarf virus RNA2 | *Bromoviridae; Ilarvirus* | NC_008037 | 97.9 | 98.1 |
| | | Senna severe yellow mosaic virus | *Alphaflexiviridae; Allexivirus* | NC_076419 | 96.6 | 97 |
| | | Mentha macluravirus 1 | *Potyviridae, Macluravirus* | OL472141 | 82.4 | 85.3 |

sampling and in addition in a "non-hop outside" pool at the 2022 sampling (Fig 3). An apple tree sample was collected in 2022 outside the hopyard 8. This individual sample as well as the hop sample in which ApMV was identified at hopyard 8 were re-analyzed by RT-PCR using the primers from Menzel et al. [31] which confirmed that ApMV was present in both the apple tree and hop plants.

## Mycoviruses and other non-plant virus sequences identified in the datasets

In addition to plant-infecting virus and viroid sequences, HTS data analysis enabled also the detection of other viral sequences associated with insects, fungi, and oomycetes. These viruses which can infect plant-associated fungi and oomycetes are known as mitoviruses or mycoviruses. In this study, several non-plant viruses were identified in the datasets (Table 7).

## Identification of other hop pathogens in the hop pool datasets

Mapping the consensus sequences of the three hop pool datasets (2021, 2022, and 2023) against NCBI BLASTn database revealed genomic sequences of hop genome as well as various sequences from other microorganisms, e.g., fungi and bacteria, some of which are known to infect hops. The presence of fungi and bacteria was confirmed by amplicon

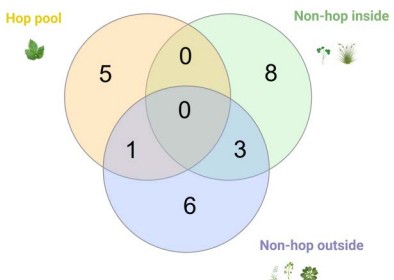

**Fig 3. Venn-diagram illustrating the viruses and viroids identified in all sample pools (hop, non-hop inside, and non-hop outside) over three years of sampling. Only one virus (ApMV) was detected in both the hop plant pool as well as in the non-hop plants outside the hopyard pool in 2022. Pasha, A. (2025)** https://BioRender.com/k13y381.

**Table 7. Non-plant viral sequences identified in the datasets.**

| Virus | Family, genus | Sequencing library (year) | Ref. genome | Pairwise ID % (BLASTn) | Pairwise ID % (BLASTx) | Natural host |
|---|---|---|---|---|---|---|
| Fusarium asiaticum mitovirus 2 | *Mitoviridae; Mitovirus* | Hopyard 4 (2022) | MZ969052 | 91.8 | 92 | *Fusarium asiaticum* |
| Plasmopara viticola lesion associated ourmia-like virus 63 | *Scleroulivirus, epsilonplasmoparae* | Hopyard 4 (2022) | ON812983 | 97.4 | 98.6 | *Plasmopara viticola* |
| Erysiphe necator associated ourmia-like virus 21 | *Penouilivirus, alphaerysiphe* | Hopyard 4 (2022) | MN611549 | 98 | 99.1 | *Erysiphe necator* |
| Hubei sobemo-like virus 34 | *Riboviria, unclassified* | Hopyard 5 (2023) | KX882879 | 80 | 84.2 | *Tetragnatha maxillosa* |
| Erysiphe necator associated ilar-like virus 1 | *Bromoviridae; Ilarvirus* | Hopyard 6 (2023) | MN630188 | 98.7 | 98.9 | *Erysiphe necator* |
| Erysiphe necator associated mitovirus 9 | *Mitoviridae; Mitovirus* | Hopyard 9 (2023) | MN557015 | 92.5 | 93 | *Erysiphe necator* |
| Alternaria arborescens mitovirus 1 | *Mitoviridae; Unuamitovirus* | Hop pool (2023) | ON714134 | 95.3 | 95.8 | *Alternaria arborescens* |

**Table 8. Hop pathogenic fungi and bacteria identified in the RNA-seq datasets and confirmed by amplicon sequencing.**

| Pathogen | Type | Disease on hop | Hop pool sample | Confirmation | |
|---|---|---|---|---|---|
| | | | | Detected by RNA-seq | Detected by amplicon sequencing |
| *Agrobacterium tumefaciens* | Bacterium | Crown gall | 2022 | Yes | Yes |
| *Alternaria alternata* | Fungus | Alternaria cone disorder | 2021, 2022, 2023 | Yes | Yes |
| *Colletotrichum sp.* | Fungus | Anthracnose | 2021, 2022, 2023 | Yes | Yes |
| *Ascochyta rabiei* | Fungus | Ascochyta leaf spot | 2021, 2022 | Yes | Yes |
| *Fusarium avenaceum* | Fungus | Cone tip blight | 2021, 2022, 2023 | Yes | Yes |
| *Phoma sp.* | Fungus | Phoma wilt | 2021, 2022, 2023 | Yes | Yes |
| *Septoria rosae* | Fungus | Septoria leaf spot | 2022 | Yes | Yes |
| *Verticillium nonalfalfae (V. albo-atrum)* | Fungus | Verticillium wilt | 2022, 2023 | Yes | Yes |
| *Verticillium dahliae* | Fungus | Verticillium wilt | 2021, 2022, 2023 | Yes | Yes |
| *Cladosporium sp.* | Fungus | Sooty mold | 2021, 2022, 2023 | Yes | Yes |
| *Fusarium sambucinum* | Fungus | Canker | 2023 | Yes | Yes |
| *Botrytis cinerea* | Fungus | Gray mold | 2023 | Yes | Yes |

sequencing. While many microorganisms (fungi and bacteria) were identified by both sequencing methods, only those previously reported to infect hop were considered in Table 8.

## Phylogenetic analysis of CBCVd

CBCVd sequences were detected only in the Hallertau region but in five different hop yards over three years of sampling. Variant analysis of the full genome sequence of CBCVd revealed eight CBCVd sequence variants, which were deposited in GenBank under accession numbers: PQ075924, PQ075925, PQ650572, PQ650573, PQ650574, PQ650575, PQ655430, and PQ655431.

The multiple sequence alignment of these eight CBCVd variants with other 87 CBCVd sequence variants obtained from GenBank showed that three of our detected variants (PQ655430, PQ650573, PQ075924) were 100% identical to three of CBCVd sequence variants previously reported in the Hallertau (PP332307, PP332297, and PP332305, respectively).

In addition, two of the variants from our study (PQ650573, PQ650572) shared 100% identity with CBCVd variants that were reported from Slovenian hops (GenBank accession numbers: KM211546 and KM211547, respectively). The phylogenetic analysis showed that the CBCVd sequence variants identified in this study were clustering with CBCVd variants from hops that were previously reported from Germany and Slovenia. This highlights their close evolutionary relationship suggesting a common origin, probably derived from a CBCVd-citrus sequence variant (Fig 4).

## Discussion

In recent years, many virome studies have been conducted to explore the viro-diversity of a crop within its ecosystem and the role of other weeds and wild plants as reservoirs of plant viruses [14–16]. In addition, a few of the studies have explored the viro-diversity at the same sampling location over two to three seasons, helping to obtain a better understanding of the viruses present in the targeted agroecosystem [14].

In this study, a preliminary HTS-based pilot study was carried out in 2021 to explore alternative reservoirs of CBCVd. Samples were collected from hops, non-hop plants growing inside and outside hopyards in locations where CBCVd was previously detected. Plant viruses may not cause symptoms on infected plants due to latency, host tolerance or just a mild strain of the virus/viroid infecting the plant [32,33]. Therefore, samples were collected regardless of whether the plants were symptomatic or asymptomatic. The pooling strategy applied in this study allowed the determination of potential hosts

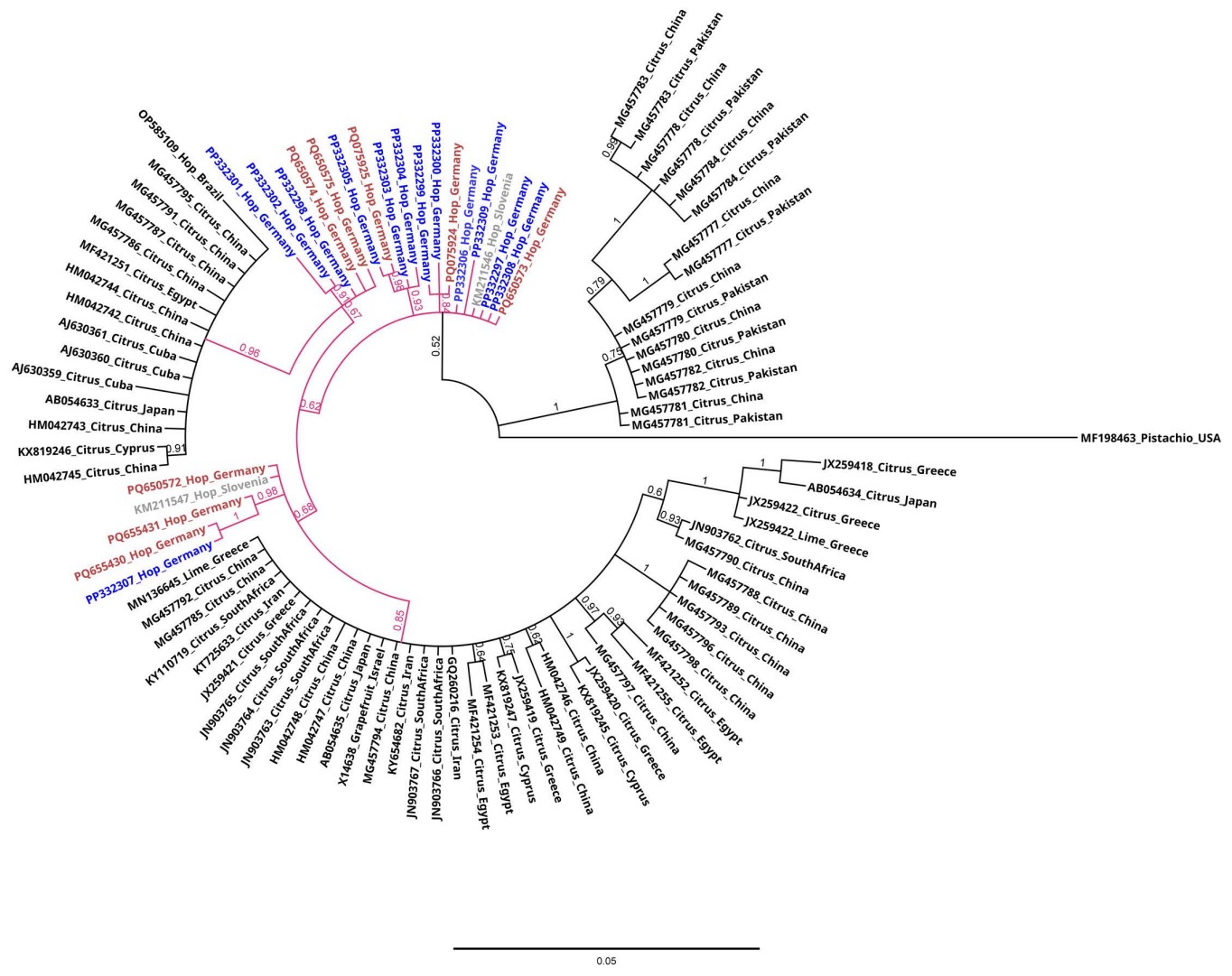

**Fig 4. Phylogenetic analysis of 95 CBCVd sequence variants.** MrBayes was used to perform Bayesian inference of phylogeny. The numbers on the branches represent Bayesian posterior probabilities indicating the statistical support for each clade. Posterior probability values range from 0.5 to 1.0. Values closer to 1.0 indicate greater support. CBCVd sequence variants from German hops are shown in blue, while other variants identified in this study are shown in red. CBCVd-hop variants from Slovenia are shown in grey. CBCVd-hop variants from Germany and Slovenia are clustered together and share the same ancestor (CBCVd-citrus). Each node is labeled with GenBank accession number of the corresponding variant, host plant, and the country where the variant was identified. The sequence of CBCVd sequence variant from pistachio (MF198463) was used as outgroup. The scale bar indicates the genetic distance.

of the detected viruses and viroids by overlapping the detections across hopyard pools with hop pool, non-hop inside and non-hop outside pools. Identified viruses in the non-hop pools were not further investigated to determine the original host plant, unless the virus/viroid was a known hop pathogen or a novel virus/viroid species. The dsRNA approach for enrichment of viral and viroid sequences enabled the detection of different viruses and viroids with varying genomic backgrounds [34]. At least ten million reads were acquired per dataset, resulting in recovering ≥ 40% of the viral genomic sequences and 100% of the viroid genomic sequences. Malapi-Wight et al. [35] found that ten million reads generated from a metagenomic sample were sufficient to obtain near-complete coverage of viral RNA and DNA genomes. In

addition, Schönegger et al. [36] found that using dsRNA as an enrichment method and 10 million reads datasets has provided more than 90% representation of a complex virome of 72 viral agents, regardless of the assembled contig lengths. The pilot study revealed the presence of known hop pathogens, in particular HpLV, HpMV, ApMV, HLVd and CBCVd (Table 3 and 4). By extending the sampling to other main hop-growing locations in Germany in the following years, it was possible to investigate any potential outbreak of CBCVd outside the Hallertau region, to get a comprehensive overview of the viruses and viroids infecting hops in Germany, and to explore other alternative host plants for the detected hop viruses and viroids. In addition, sampling from the three main hop-growing locations over two consecutive years targeting the same hopyards helped to investigate spatio-temporal differences at these regions which may have been overlooked if only one region had been sampled for one year. Sampling in the Hallertau in 2022 and 2023 targeted different hopyards than those sampled in 2021, as the three hopyards were eradicated following the severe CBCVd outbreaks.

Interestingly, the total number of viruses and viroids detected in this study is less than in other virome studies. For example, Rivarez at al. [15] detected 125 viruses, including 79 new species in the Slovenian tomato virome study. Similarly, Gaafar et al. [14] reported 35 viruses in addition to nine associated nucleic acids when they looked at the pea virome in Germany. It is counter intuitive that hop, being perennial with very few replacements of plants, is a reservoir to fewer viruses and viroids than annual crops such as tomatoes or peas. In addition, hop planting material in Germany has been propagated through certified virus-free production schemes in dedicated nurseries for more than three decades. These efforts align with the EPPO standard "Certification Scheme for Hop" [37], which aims to minimize the spread of viral agents and other pests through planting material. These phytosanitary measures likely contribute significantly to the reduced pathogen diversity observed in German hopyards. Also, it may be possible that due to the sampling and pooling strategy that the amount of plant material may have been insufficient to detect low titer viruses and viroids in the pooled samples. Although the pools (already representing between 30–210 individual plant samples) contain at least 6 g of plant material, only a fraction of this could be used for dsRNA enrichment and HTS (Table 2).

## Common viruses and viroids in German hops

In total, four viruses and two viroids were detected in all hop pools over the three years. Two of them (HpLV and HpMV) were detected across all the nine hopyards (Table 3 and S6 Table), which means that both carlaviruses are widely distributed in German hopyards. Both are mainly spread between hop plants by aphid vectors in a non-persistent manner and mechanical means [2]. HpLV and HpMV have been shown to occur in all hop-growing locations in the world [3]. HpLV infection tends to be symptomless. In contrast, HpMV is associated with leaf mosaic symptoms, stunted growth, and lower cone yield in some old hop cultivars. However, it is often asymptomatic in most commercial hop cultivars grown today [2]. In the United States, mixed infection of HpLV and HpMV in the hop cultivar Chinook caused significant impact on lateral length, leaf weight, cone yield, number of cones per plant, and reduction of the organic acids and oils content in infected plants [38].

ApMV belongs to the genus *Ilarvirus* and was detected in one hopyard in 2021, in three hopyards in 2022, and in five hopyards in 2023 (S6 Table). Interestingly, in two hopyards, ApMV was not detected in 2022 but was detected one year later. This indicates either the infected plants were missed during sampling or only few plants were infected in 2022 but the infection spread to more plants in 2023 thus allowing to be collected during sampling. As the sampling was carried out randomly in each year, this question cannot be answered. Interestingly, ApMV sequences of all three viral RNAs were identified in the non-hop plants pool from outside a hopyard in 2022. ApMV was confirmed by RT-PCR in the same pool and in an individual sample belonging to the same pool, collected from an apple tree outside hopyard 8 (Elbe-Saale, Dresden). It may be possible that ApMV was transmitted to the hop plants when apple harvest residues were used as fertilizer in the hopyard, as ApMV is transmitted exclusively by mechanical means [39]. This hypothesis is based on our observations that at some of the sampling sites, apple harvest residues are used in hopyards as organic fertilizer. It is also possible that the growers may have used uncertified planting material that could have been infected with ApMV and/

or other hop pathogens. ApMV is considered one of the most harmful viruses for hop production. The spread of ApMV in hopyard is slow but it is strongly dependent on the cultivar [2]. ApMV seems to reduce the ability to propagate hop plants vegetatively, and it can also reduce alpha acid content by up to 22% [2,40]. The detection of ApMV along with HpMV and HpLV in many hopyards across different hop growing locations in Germany may be alarming, as the co-infection of these viruses can cause a significant reduction in the cone weight and alpha acid content. This reduction is attributed to the low number of lupulin glands in the cones of co-infected hop plants [2,40].

The detection of ArMV in one hopyard (Tettnang) and only in one sampling (2022) may indicates that this virus is less common in the German hops. ArMV infection has been associated with several diseases including spidery hop, split leaf blotch, and hop chlorotic disease [2]. Significant yield losses were demonstrated in New Zealand [41] when hops were co-infected with ArMV with HpMV or HpLV despite the absence of symptoms. However, nematodes transmitting ArMV are less mobile between plant roots, so their effect is locally limited [42].

HLVd was detected in the nine hopyard pools over the three years, confirming that it is widely distributed in commercial hops worldwide [5]. HLVd causes often latent infection in the majority of hop cultivars and was firstly reported in German hops by Puchta et al. [43]. HLVd infection is less severe than other viruses or viroids infecting hops, but can still result in reduction of alpha acid content [44]. Our findings align with those from Eppler [42], who investigated the viro-diversity in many hop growing regions in Germany and tested the collected samples by ELISA for 15 viruses which were previously reported in hops, e.g., HpLV, HpMV, ApMV, ArMV, AHLV, potato virus X (PVX), potato virus Y (PVY), tobacco mosaic virus (TMV), cucumber mosaic virus (CMV), etc. The results showed that there was no hop growing region or hop variety virus-free. HpLV, HpMV, ArMV, and ApMV were detected in the tested samples. Furthermore, dot-blot hybridization was used to detect viroids. HLVd was detected in many of the tested samples, whereas HSVd, though present in Germany in grapevine since the 1980s [45], was not detected in any of them. In addition, during the monitoring of hop viruses in Germany from 2008 to 2012, Seigner et al. [46] analyzed 1,170 symptomatic hop leaf samples collected from all German hop growing regions by ELISA and RT-PCR. The results showed that HpLV, HpMV and ApMV were the omnipresent viruses in the German hops, while ArMV was rarely detected, which is also in line with the findings of our study. In addition, HSVd was detected in nine samples in 2009, and the infected plants were eradicated effectively.

Although CBCVd infection is still limited to the Hallertau region, as confirmed by our results, it could be possible that CBCVd infection is spreading to new hopyards in the same region. This is shown by the data from hopyard 4 (Hallertau), where CBCVd was not detected in 2022, but was detected in 2023. These results are consistent with the findings of the LfL during their CBCVd-monitoring programme [24]. There could also be the risk of spread to other regions in the future, if no sufficient measures are applied. Currently, CBCVd is regulated in the European Union as a regulated non-quarantine pest (RNQP), which means that the threshold for planting material (other than pollen and seeds) of *Humulus lupulus* is 0%. Furthermore, several requirements for the production of hop plants for planting are mandatory (http://data.europa.eu/eli/reg_impl/2019/2072/2025-01-26, accessed on May 15. 2025).

Štajner et al. [47] found that coinfected hop plants with HLVd and CBCVd showed a significant reduction in biomass up to 81% compared to plants infected with either HLVd or CBCVd. These findings highlighted the aggressiveness of CBCVd in causing a severe disease in hops with synergistic interaction with HLVd.

## Identified viruses in alternative plants inside and outside the hopyards

We were interested to know whether we could identify alternative hosts for the viruses and viroids found in the hop plants. In particular, we wanted to know if there were alternative hosts for CBCVd as this would have had implications on control measures. Several virome studies identified alternative reservoirs for viruses infecting a crop, e.g., in the case of tomatoes where ten viruses were detected in both the tomato and the surrounding weeds [15] or peas where two pea viruses were also identified in non-legume weeds [14].

For this purpose, many samples were collected from weeds and wild plants growing inside and outside the hopyard. The number of the collected samples from inside the hopyard (five per yard) was lower than the collected samples from the outside (ten per yard, Fig 1) because the weeds were removed from the hopyard as agricultural practice during hop production. Furthermore, regular tilling and producing "hop hills" in which the hop plants were growing, were also seen as common practice thus reducing the occurrence of weeds within the crops. Many of the hopyards that were sampled were surrounded by other hopyards, in particular in the Hallertau region; in the case of Tettnang region, apple orchards were often seen next to hopyards.

All the viruses that were identified in non-hop inside and non-hop outside pools (except ApMV), are not known to infect hop. Eppler [48] investigated for alternative reservoirs for HpMV and ArMV in Tettnang. Samples of 36 weed species belonging to 10 plant families were collected from different hopyards, and tested by ELISA techniques. ArMV was detected in 22 plant species, while HpMV was detected in four plant species. However, the results of ArMV detection in 15 of the tested species, and same for HpMV detection in 2 of the tested species, were questionable due to weak ELISA reaction. Further attempts to induce infections in hops by mechanical inoculation of these potentially infected weed species were unsuccessful. However, in our study, none of the reported hop viruses and viroids were detected in non-hop plant pools, except ApMV.

From the pools of non-hop plants within the hopyards, we detected viruses of weeds and root vegetables. Other viruses detected in non-hop outside pools are known to infect weeds and other crop plants including legumes and fruit trees. Two of these viruses are economically important: peanut stunt virus (PSV) and prune dwarf virus (PDV). PSV was detected in a white clover (*Trifolium repens*) sample collected from Tettnang (non-hop outside pool, 2023). PSV was reported to artificially infect a wide range of legume species, and is known to naturally infect peanut (*Arachis hypogaea*) [49]. Since the climatic conditions in Germany are not suitable for peanut cultivation, we do not know how this virus arrived in Germany. However, PSV may pose a risk to other legume species cultivated in Germany. PDV is known to infect stone fruits and causes significant losses. PDV was detected in a European pear (*Pyrus communis*) sample collected from Elbe-Saale region (Halle) (non-hop outside pool, 2023).

### Mycoviruses and non-plant viruses detected in the datasets

The implementation of HTS in virome studies has revealed not only plant-associated viruses and viroids, but also viruses of other organisms that live on plants, such as fungi, oomycetes, and insects. Although mycoviruses are generally not considered pathogenic to plants, their association with pathogenic fungi suggests that they may influence the dynamics of fungal infections in plants, and thus they may play a role as biocontrol agents of fungal infections [50]. For instance, Al Rwahnih et al. [51] showed in their study that the grapevine virome is dominated by mycoviruses. In another study, nine mycoviruses were identified by Vinogradova et al. [52] in their investigation of the mycovirome of the vineyard.

We identified six mycovirus sequences present in different sample pools (Table 7). Among them, alternaria arborescens mitovirus 1 was detected in a hop pool sample from 2023. The host sequence (*Alternaria arborescens*) was also detected in the same sample pool (Table 8). This virus was also detected before in a virome study characterizing the mycovirome of the vineyard [52]. Its fungal host *Alternaria arborescens* was reported on hops [53]. Their study suggested that the fungal pathogen is latent on the leaf surface and is only pathogenic when the host is injured. Other mycoviruses were identified in hop yard pools containing both hop and non-hop plants. These mycoviruses were associated with fungal or oomycete hosts that have not previously been reported to infect hops. However, further investigation is needed. A non-plant virus was detected (Hubei sobemo-like virus 34). It is associated with stretch spiders (*Tetragnatha maxillosa*), which are one of the main components of the natural enemies of crop pests in ecosystems [54].

### Identification of other hop pathogens in the hop pool datasets

The reuse of HTS datasets is becoming an important topic in life sciences, particularly in the frame of open science and fair data sharing. Although data reuse has its limitations and challenges, it can successfully lead to novel discoveries [55]. In plant pathology, re-analysis of previously published RNA sequencing datasets, originally used to detect viruses and

viroids, allowed the detection of other plant-associated pathogens [56]. In addition, the use of the virus and viroid enrichment methods does not fully eliminate RNAs of the host plant and other microorganisms living on the surface of the plant (phyllosphere), within the plant (endosphere), as well as from insects and nematodes that have physical contact with the plant [57].

In our study, we expanded the scope of the virus and viroid focused analysis by mapping consensus sequences from the three hop pool datasets (2021, 2022, and 2023) against the NCBI BLASTn database. This approach revealed hop genome sequences as well as sequences of other microorganisms, e.g., fungi and bacteria. These microorganisms may be epiphytic, living on the plant surface, non-harmful to plants, or may have been introduced through environmental contamination, making it challenging to determine true hop pathogens. By cross-referencing the identified fungal and bacterial species with previous literature [3,53,58] we were able to identify fungi and bacteria which are pathogenic to hops. One bacterium sequence (*Agrobacterium tumefaciens*) was identified in the dataset of hop pool 2022. This bacterium was previously reported in hops [58]. In two pools from 2022 and 2023, *Verticillium nonalfalfae* (formerly *V. albo-atrum*) sequences were identified*; V. nonalfalfae* causes a severe wilt disease threating the hop production in Europe [59]. However, other pathogenic hop fungal species detected seem to be a low risk to hop production as none of them are known to cause severe diseases on hops.

Although this approach is not yet optimized to eliminate potential false positive results caused by cross contamination during sampling or sample preparation for sequencing, it highlights the importance of expanding the use of the virome datasets to unveil other plant-associated microorganisms. It may be an interesting topic for future research to achieve a more comprehensive understanding of plant health, including pathogen-pathogen and pathogen-plant interactions.

## Phylogenetic analysis of CBCVd

The phylogenetic analysis of CBCVd sequences generated in this study and from sequences obtained from GenBank revealed that CBCVd isolates from hop samples share the same ancestor with CBCVd isolates found in citrus hosts with nucleotide identities ranging from 84.9% to 95.8%. These findings are in line with the Slovenian hypothesis [60] that the CBCVd outbreak in Slovenia was associated with the establishment of a hopyard on a former waste dump of citrus fruit residues. In addition, the phylogenetic analysis showed that two of the CBCVd variants identified in this study are 100% identical to the CBCVd variants from Slovenia, while other CBCVd sequence variants from Germany clustered closely with the Slovenian variants which may indicate the illegal trade of CBCVd-infected, vegetative hop material from Slovenia.

## Conclusion

This study presents a comprehensive investigation of the diversity and prevalence of viruses and viroids infecting German hops. Although the sampling targeted a small number of hopyards compared to the total cultivated hop area in Germany, the results present an overview of the dominant viruses and viroids in the German hopyards. Surprisingly, in contrast to other virome studies, a smaller number of viruses and viroids were identified in hop samples. Furthermore, apart from ApMV, no common host reservoirs between hop viruses and alternative host plants were identified. However, the hop viruses and viroids identified in this study using HTS were similar to those detected in a previous study conducted more than 30 years ago using ELISA as a diagnostic tool [42]. Since then, only CBCVd was a novel pathogen infecting German hops in the Hallertau region, as it was detected for the first time in 2019. It is interesting to note that *H. lupulus* does not seem to be very susceptible to viral infections when compared to other crops such as tomatoes (*Solanum lycopersicum*), potatoes (*Solanum tuberosum*) or legumes (e.g., *Vicia faba*, *Pisum sativum, Phaseolus vulgaris*). However, the host change from citrus plants to hops had a severe effect on this new host plant with hop plants dying within a few years of infection. Therefore, the possibility that other viruses or viroids from different hosts may have a similar dramatic effect on hops cannot be excluded. HTS analyses of propagation material may help to provide virus- and viroid free plantlets to establish healthy production sites. The detection of fungal and bacterial hop sequences in the virome datasets also

highlights the broader usefulness of HTS beyond virus discovery, including the improved understanding of interactions of organisms (within and between species) due to the high detection capacity, providing deeper insights into plant diseases and an integrated analysis for plant health.

## Supporting information

**S1 Table. Samples collected in 2021.** Description of all plant samples collected from three hopyards, including the sampling location, sample type (hop or non-hop), taxonomic classification, and observed symptoms.
(XLSX)

**S2 Table. Samples collected in 2022.** Description of all plant samples collected from six hopyards including the sampling location, sample type (hop or non-hop), taxonomic classification, and observed symptoms.
(XLSX)

**S3 Table. Samples collected in 2023**. Description of all plant samples collected from six hopyards including the sampling location, sample type (hop or non-hop), taxonomic classification, and observed symptoms.
(XLSX)

**S4 Table. Numbers of obtained raw-reads, sequences left after QC including removal of duplicate reads and numbers of assembled contigs from QC-controlled reads.**
(XLSX)

**S5 Table. Primers used for conventional (RT-)PCR confirmation.**
(XLSX)

**S6 Table. Viruses and viroids identified in each sequence data pool.**
(XLSX)

## Acknowledgments

The authors would like to thank Jonas Hartrick (deceased) and Hannes Piepenbrink for the technical assistance. The authors would like to express their gratitude to the LfL, Institute for Crop Science and Plant Breeding, IPZ 5b Plant Protection, for their support and the assistance with the sampling in the Hallertau over the three years. We are grateful to HVG (Hopfenverwertungsgenossenschaft e. G.) for their support and partial funding of the sampling trip.

We would like to thank Max Weber (Hopfenversuchsanlage Strass, LTZ) for allowing us the access to their hopyards and for helping with sampling in 2022 and 2023. We would like to extend our thanks to APH e.G. Hinsdorf GbR and Taucherwald Agrar GmbH for allowing the sampling in their hopyards in Elbe-Saale region.

We are grateful to Dr. Michael Hagemann, University of Hohenheim as well as Dr. Sebastjan Radišek, Slovenian Institute of Hop Research and Brewing, for the fruitful cooperation and discussion.

We would like to extend our gratitude to Prof. Dr. Boas Pucker, University of Bonn, for the constructive comments on the manuscript.

## Author contributions

**Conceptualization:** Ali Pasha, Gritta Schrader, Heiko Ziebell.

**Data curation:** Ali Pasha.

**Formal analysis:** Ali Pasha.

**Funding acquisition:** Gritta Schrader, Heiko Ziebell.

**Investigation:** Ali Pasha.

**Methodology:** Ali Pasha, Heiko Ziebell.

**Project administration:** Gritta Schrader, Heiko Ziebell.

**Software:** Ali Pasha.

**Supervision:** Gritta Schrader, Heiko Ziebell.

**Validation:** Ali Pasha.

**Visualization:** Ali Pasha.

**Writing – original draft:** Ali Pasha.

**Writing – review & editing:** Ali Pasha, Gritta Schrader, Heiko Ziebell.

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
