## [Decision Letter · Decision Letter 0]

Dear Dr. Ziebell,

Thank you for submitting your manuscript to PLOS ONE. After careful consideration, we feel that it has merit but does not fully meet PLOS ONE’s publication criteria as it currently stands. Therefore, we invite you to submit a revised version of the manuscript that addresses the points raised during the review process.

Both reviewers gave detailed revisions on your manuscript. I hope you can complete the revision fast.

We look forward to receiving your revised manuscript.

Kind regards,

Ahmet Uludag, Ph.D.

Academic Editor

PLOS ONE

Journal Requirements:

“The project was supported by funds of the Federal Ministry of Food and Agriculture (BMEL) based on a decision of the Parliament of the Federal Republic of Germany via the Federal Office for Agriculture and Food (BLE) under the innovation support programme (grant number 2818714B19).

The sequencing was funded by the Euphresco projects: VirusCurate 2019-E-312: Using High Throughput Sequencing to gain insights from virus collections and strengthening the infrastructure of Plant Virus Collections and 2020-A-347: Baseline studies on virus reservoirs. “

Reviewers' comments:

Reviewer's Responses to Questions

**Comments to the Author**

1. Is the manuscript technically sound, and do the data support the conclusions?

Reviewer #1: Yes

Reviewer #2: Yes

2. Has the statistical analysis been performed appropriately and rigorously?

Reviewer #1: Yes

Reviewer #2: Yes

3. Have the authors made all data underlying the findings in their manuscript fully available?

Reviewer #1: Yes

Reviewer #2: Yes

4. Is the manuscript presented in an intelligible fashion and written in standard English?

Reviewer #1: Yes

Reviewer #2: Yes

Reviewer #1: The manuscript “Virus and viroid diversity in hops, investigating the German hop virome” submitted by Dr. Ziebell presents comprehensive and well-structured study on the Hop virome in different hop growing areas in Germany. The study provides valuable insights into the virome of German hops using high-throughput sequencing over multiple years and regions.

While the manuscript is scientifically sound and methodologically robust, there are several areas where clarification, additional detail, or minor revisions would enhance the clarity and impact of the work. These are outlined below:

Lines 52-55: Please include the full names of all viruses and viroids mentioned, even if they were previously introduced in the abstract. This ensures clarity and consistency for readers who may refer to specific sections independently of the abstract.

Materials and methods:

For sampling in 2022 and 2023 add the information which hopyard was collected in which year(s). In Table 1 you have the data about temperature on sampling day, then you never refer to this data anymore. It can be moved to some supplementary table and instead the data regarding the exact year of collection should be added.

All supplementary tables should have titles in the beginning, or it can be the first sheet of the table. Now they are in the end of the manuscript, but it would be easier to reader is the titles aro also in the tables.

Supplementary S2 Table. In some tables you have “Sample No.” information in some you don’t have it. Suggest unifying.

Did you perform any search for potentially new viruses/viroid which are more divergent from the RefSeq sequences. If I am not mistaken mapping parameters of “High sensitivity/Medium” are something 80–85% regarding similarity of sequences. What about potentially more divergent sequences? Suggest Diamond BlastX or pFam search.

Which controls you included in the sequencing. Was all sequencing performed in one lane? How are you sure that the result where you have “Sequencing depth” low or very low is not cross talk?

Lines 171-173 you wrote that you designed primers for identified viruses, then in S4 you have some primers from literature (with references). Please, be clear about this in the text and include full references in the S4.

In S5 Table you have a column “Seq depth” is it a “average coverage” at one place you describe it, but it should be explained in each instance, and you also use sequencing dept od 150bp, so it can be confusing for potential reader. Did you apply any threshold when creating final consensus sequences? Instead of “Sample” it should be “Sequencing library”

I suggest creating new Supplementary table where you will for all sequenced libraries collect following data (from table S5) “Library read length” “Total raw reads” and “Reads no. after QC” and you can include year f sampling (in that way you will address one of my previous comments).

Results:

In line 44 of S5 you have for “Total raw reads” and “Reads no. after QC” numbers which are probably belonging to other library or the result is from another library?

In table 3 you have most data repeated from S5. I suggest removing most of the repeated data and include the results of Blast (nucleotide and amino acid-where applicable) for the consensus sequence.

In Table 3 for virodid you have more than one accession. Describe very clearly how you obtained those sequences (how you called variants how you are sure that those are not “artificially” assembled sequences). I understand that they are most probably consequence of pooling samples, but you need to be very clear about how your decisions were made.

Blast results of consensus sequences should be added Table 6.

In table 7 add the column of the pathogen detected in the data (if possible). For example you have Verticilium wilt 2 times with different years, are they different Verticilium species?

Unify writing of GebBank (sometimes it is written Genbank).

In figure 4 caption indicate what number on branch represents.

Discussion:

The discussion part can be shortened.

Regarding ApMV can the detection in hop be consequence of pollen contamination?

Reviewer #2: The manuscript PONE-D-25-26477, entitled "Virus and viroid diversity in hops: Investigating the German hop virome", presents an important study on the prevalence of viruses and viroids in hop production areas in Germany. It also explores potential reservoirs of the viroid CBCVd, which was recently detected in hops and is known to cause severe damage to production. The analysis is based on high-throughput sequencing (HTS), and the resulting data were also used to detect the presence of mycoviruses and other hop pathogens in the collected samples. From an epidemiological perspective, the phylogenetic analysis of CBCVd is particularly noteworthy. It demonstrates that sequences from hop samples share high identity with variants previously identified in citrus plants, supporting earlier hypotheses about cross-host transmission from citrus fruits. The results highlight the importance of producing and trading healthy planting material. Since hop cultivation relies on vegetative propagation, infected material can readily transmit dangerous pathogens. Before publishing I recommend few minor corrections:

Page 5, Line 103: Please clarify what is meant by "outside the hop yard." What was the approximate distance from the nearest hop yard? Additionally, why were wild hop plants—which often grow in the vicinity of hop yards—not included in the sampling?

Page 10, Lines 210–215 and Table 3: The results for the “hop pool” are presented and discussed. I suggest including the data for the “hop yard pool” as well, to allow for comparison and discussion of samples originating from different hop-growing regions.

Page 14, Table 7: Please add a column listing the Latin names of the fungi and bacteria identified in the RNA-seq dataset. Consider removing the reference column, as the cited sources do not represent first reports of the listed pathogens.

Page 18, Lines 329–331: Please consider adding that the lower diversity may also be attributed to the fact that, for more than three decades, hop planting material in Germany has been propagated through certified production of virus-free material in hop nurseries and virus-free hop yards, in accordance with the EPPO standard PM 4/16 (2) – Certification Scheme for Hop.

Page 19, Line 362: correct to uncertified planting material.

Page 19, Line 362-368: ApMV is the most widespread and economically important ilarvirus affecting hop. Numerous studies conducted in the 1990s in the UK, USA, Czech Republic, Poland, and Germany demonstrated its negative impact on hop yield and quality. Several studies have also shown a synergistic effect between ApMV and other hop viruses. These findings prompted the implementation of virus-free hop planting material production in most European hop-growing countries. I suggest rewriting this section to include these points, supported by relevant references (see, for example, Pethybridge et al., 2008).

Page 20, Lines 388–392, Reference 45: Please add that during the monitoring period (2008–2012), HSVd was detected in a few hop plants, which were subsequently eradicated.

Page 20, Line 399: planting material

Page 21, Lines 420: “Earth dams” is not correct term. Probably you mean hop hilling.

Page 21, Line 423: add what kind of orchards. Apple?

Page 21, Line 425-427: Apple plants are not host of Phorodon humuli. The winter hosts are Prunus sp.

Page 24, Line 489: Agrobacterium tumefaciens, and has been reported on nearly every country where hops are commercially grown.

Page 24, Line 490: Verticillium albo-atrum. The species on hop is now classified as V. nonalfalfae.

**Do you want your identity to be public for this peer review?** For information about this choice, including consent withdrawal, please see our Privacy Policy

Reviewer #1: No

Reviewer #2: No

---

## [Author Response · Author response to Decision Letter 1]

3 Jul 2025

We would like to thank the academic editor and both reviewers for their valuable feedback and for evaluating our manuscript. We have revised the manuscript accordingly and provided detailed responses to each comment below. All changes are marked with track changes in the revised manuscript.

All supplementary tables have been updated according to the reviewers’ suggestions.

All tables (except table 2) have been updated according to the reviewers’ suggestions.

Reviewer #1: The manuscript “Virus and viroid diversity in hops, investigating the German hop virome” submitted by Dr. Ziebell presents comprehensive and well-structured study on the Hop virome in different hop growing areas in Germany. The study provides valuable insights into the virome of German hops using high-throughput sequencing over multiple years and regions.

While the manuscript is scientifically sound and methodologically robust, there are several areas where clarification, additional detail, or minor revisions would enhance the clarity and impact of the work.

These are outlined below:

Lines 52-55: Please include the full names of all viruses and viroids mentioned, even if they were previously introduced in the abstract. This ensures clarity and consistency for readers who may refer to specific sections independently of the abstract.

Thank you for the suggestion. It has been addressed.

Materials and methods:

For sampling in 2022 and 2023 add the information which hopyard was collected in which year(s). In Table 1 you have the data about temperature on sampling day, then you never refer to this data anymore. It can be moved to some supplementary table and instead the data regarding the exact year of collection should be added.

Samples were collected from the same hopyards in 2022 and 2023, as shown in Table 1, marked with “X”. An explanation has been added to the heading of Table 1.

Table 1 has been updated, and the temperature information has been moved to Supplementary Tables 1, 2, and 3.

All supplementary tables should have titles in the beginning, or it can be the first sheet of the table. Now they are in the end of the manuscript, but it would be easier to reader is the titles aro also in the tables.

The title of each supplementary table has been added in the first sheet of the Excel file.

Supplementary S2 Table. In some tables you have “Sample No.” information in some you don’t have it. Suggest unifying.

We unified the format by removing the “Sample No.” column because it refers to the sample number in our internal database.

Did you perform any search for potentially new viruses/viroid which are more divergent from the RefSeq sequences.

If I am not mistaken mapping parameters of “High sensitivity/Medium” are something 80–85% regarding similarity of sequences.

What about potentially more divergent sequences? Suggest Diamond BlastX or pFam search.

Thank you for this valuable comment. Yes, we explored the possibility of detecting novel viruses or viroids. We further analyzed consensus sequences derived from de novo assembly that showed any similarity to viral or viroid sequences using both BLASTn and BLASTx against the NCBI non-redundant database to identify potential matches. We also performed additional BLASTn searches against the broader NCBI nucleotide database for sequences with low pairwise identity to known viruses or viroids in RefSeq. However, these sequences were highly similar to known plant-derived sequences or to other microorganisms present in the sample.

As mentioned in the M&M, the “High sensitivity/Medium” setting was used to re-map the raw reads to the selected reference genomes, ensuring accurate read abundance quantification. This step followed the identification and taxonomic assignment of candidate viral sequences and was not the primary method used for discovering novel viruses. However, no novel viral or viroid species were identified in this study.

We focused on the identified hop virus and viroid sequences from the German hopyards, all of which were submitted to GenBank (see Table 3).

Which controls you included in the sequencing.

Was all sequencing performed in one lane?

How are you sure that the result where you have “Sequencing depth” low or very low is not cross talk?

Thanks for this important comment. Although internal (spiked) control was not included in the sequencing libraries, several quality indicators confirm the efficiency and reliability of our sequencing runs.

The target for each dataset was 10 million reads. In practice, we obtained at least 12 million raw paired-end reads per dataset (S4 Table).

Furthermore, the successful mapping of more than 100 reads to small genomes, such as viroids, indicates that the libraries were suitable for downstream analysis (S6 Table).

Consequently, these results support the efficiency of the sequencing despite the absence of an internal control. However, we acknowledge the importance of including internal controls in sequencing experiments and will use them in future studies to strengthen quality of the results.

According to the service provider, samples from the same year were sequenced in the same lane unless a sample had to be resubmitted due to an initially low RNA concentration.

We acknowledge that low-level cross-talk between samples cannot be fully excluded during library preparation or sequencing. To address this issue, we performed RT-PCR using RNA extracted from each sample pool to confirm the presence or absence of the identified virus or viroid. This validation step helped us distinguish between true low-titer infections and potential index cross-talk.

We also recognize that low sequencing depth may indicate a low viral or viroid titer in the sample. For this reason, we did not apply a strict cut-off threshold for read depth in our detection criteria. However, since sequencing depth per virus or viroid was not directly relevant to our study’s main objective of detecting “presence or absence” rather than performing quantitative transcriptomics, we have removed “Seq depth” from S6 Table.

Since not all viral genomes were fully recovered, we changed the “Seq depth” column in S6 Table to “average coverage”.

Lines 171-173 you wrote that you designed primers for identified viruses, then in S4 you have some primers from literature (with references). Please, be clear about this in the text and include full references in the S4.

In lines 171-173 we are referring to the tool used for primer design. A clarification has been added in lines 204-205.

In S5 Table you have a column “Seq depth” is it a “average coverage” at one place you describe it, but it should be explained in each instance, and you also use sequencing dept od 150bp, so it can be confusing for potential reader. Did you apply any threshold when creating final consensus sequences? Instead of “Sample” it should be “Sequencing library”

Thanks for the correction. As mentioned above that the “Seq depth” column has been substituted with “Average coverage”. The recovered length (bp) and (%) of each identified virus and viroid genomes have been added to S6 Table.

At least 40% of the virus genomes and 100% of the viroid genomes were recovered and confirmed by another diagnostic test to give the results of this study more reliability.

I suggest creating new Supplementary table where you will for all sequenced libraries collect following data (from table S5) “Library read length” “Total raw reads” and “Reads no. after QC” and you can include year f sampling (in that way you will address one of my previous comments).

Thanks for this suggestion. A new supplementary table (S4) has been created to describe the content of each Bioproject of RNA-seq.

The information related to the sequencing libraries has been removed from S6 Table to avoid redundancy.

Results:

In line 44 of S5 you have for “Total raw reads” and “Reads no. after QC” numbers which are probably belonging to other library or the result is from another library?

It has been corrected. The “Total raw reads” and “Reads no. after QC” numbers have been moved to the S4 Table.

In table 3 you have most data repeated from S5. I suggest removing most of the repeated data and include the results of Blast (nucleotide and amino acid-where applicable) for the consensus sequence.

To avoid repetition: the genome recovery has been moved to S6 Table, and BLASTn and BLASTx results have been added to Tables 5, 6, and 7.

In addition, to make the results easier to understand and compare, so we have created a new table (Table 3) presenting all detected viruses and viroids in all sample pools over the three years.

In Table 3 for virodid you have more than one accession. Describe very clearly how you obtained those sequences (how you called variants how you are sure that those are not “artificially” assembled sequences). I understand that they are most probably consequence of pooling samples, but you need to be very clear about how your decisions were made.

Thank you for mentioning this important point. We focused particularly on CBCVd diversity because this viroid recently emerged in German hop cultivation and is known to cause significant economic losses in European hop production. The observed sequence variation among CBCVd isolates supports the hypothesis that the viroid jumped from infected citrus plants to hops.

We submitted multiple viroid sequences to GenBank based on the identification of sequence variants within pooled samples. These variants were identified after de novo assembly and taxonomic identification. They were further validated through reference-guided mapping. This ensured that the identified sequences were true variants and not artifacts of sequencing or misassembly.

The following criteria were considered:

1. Sequence variants were only considered if the coverage depth at the site of variation was ≥5 reads, which reduces the likelihood that sequencing errors will be mistaken for real variants.

2. Each variant sequence was identified at least in one distinct sample pool (i.e., from different hopyards or hop pools).

3. We identified and verified variants using the Variation/SNPs tool in Geneious Prime, which allowed us to visualize and manually confirm SNPs across the viroid genomes.

Due to the use of pooled RNA samples and the known high mutation rates of viroids (Flores et al., 2014), CBCVd and HLVd quasispecies were expected.

The predominance of CBCVd sequences that were identical to or highly similar to Slovenian isolates supported the reliability of our methodology for variant analysis.

The analysis details have been added to M&M. In addition, a clarification has been added as a footnote to Table 3.

Blast results of consensus sequences should be added Table 6.

It has been added to the respective Tables.

In table 7 add the column of the pathogen detected in the data (if possible). For example you have Verticilium wilt 2 times with different years, are they different Verticilium species?

A column of “Pathogen” containing the Latin names of detected pathogen has been added to the Table 7.

Unify writing of GebBank (sometimes it is written Genbank).

It has been unified “GenBank”.

In figure 4 caption indicate what number on branch represents.

Thanks for the suggestion. The information has been added to the caption.

Discussion:

The discussion part can be shortened.

We acknowledge the reviewer’s suggestion to shorten the discussion section. After carefully reviewing the section, we made minor edits to improve clarity and conciseness. However, we believe that a detailed discussion of our findings and a comparison with previous studies is necessary to provide context for the results and emphasize the importance of our work. Therefore, we have retained the structured format, beginning with an overview of the study and its main findings and then providing details on the detected pathogens and their potential impacts on hops.

Regarding ApMV can the detection in hop be consequence of pollen contamination?

In general, fruit trees are flowering in Germany in April/May. Our sampling time was at the end of June/beginning of July so we think it is very unlikely that ApMV-contaminated apple pollen was still present as residual surface contaminant on our hop samples.

Reviewer #2: The manuscript PONE-D-25-26477, entitled "Virus and viroid diversity in hops: Investigating the German hop virome", presents an important study on the prevalence of viruses and viroids in hop production areas in Germany. It also explores potential reservoirs of the viroid CBCVd, which was recently detected in hops and is known to cause severe damage to production. The analysis is based on high-throughput sequencing (HTS), and the resulting data were also used to detect the presence of mycoviruses and other hop pathogens in the collected samples. From an epidemiological perspective, the phylogenetic analysis of CBCVd is particularly noteworthy. It demonstrates that sequences from hop samples share high identity with variants previously identified in citrus plants, supporting earlier hypotheses about cross-host transmission from citrus fruits. The results highlight the importance of producing and trading healthy planting material. Since hop cultivation relies on vegetative propagation, infected material can readily transmit dangerous pathogens. Before publishing I recommend few minor corrections:

Page 5, Line 103: Please clarify what is meant by "outside the hop yard." What was the approximate distance from the nearest hop yard? Additionally, why were wild hop plants—which often grow in the vicinity of hop yards—not included in the sampling?

A clarification has been added to “outside the hopyard”. The samples were randomly collected in small proximity to the hopyard itself, approximately not exceeding 50 meters. Wild hop plants were not present near the hopyards; due to the risk imposed by male wild hop plants, any wild growing hops is generally eradicated to prevent pollination of the production site.

Page 10, Lines 210–215 and Table 3: The results for the “hop pool” are presented and discussed. I suggest including the data for the “hop yard pool” as well, to allow for comparison and discussion of samples originating from different hop-growing regions.

Thank you for your suggestion. The detailed analysis results for all hopyard pools are listed in S6 Table. However, we created a new table (Table 3) which shows all the viruses and viroids detected in the hopyard pools, as well as in the hop and non-hop (inside and outside) pools over the three sampling years.

Page 14, Table 7: Please add a column listing the Latin names of the fungi and bacteria identified in the RNA-seq dataset. Consider removing the reference column, as the cited sources do not represent first reports of the listed pathogens.

A new column “Pathogen” has been added, containing the Latin names of the detected bacteria and fungi. The reference list has been removed, and the references are now cited in the discussion section.

We agree that these references were not the first reports of these pathogens. However, they contain valuable information about pathogens that infect hops. We used these references to overlap the RNA-seq analysis results with known hop pathogens in order to determine which of the identified pathogens infect hops.

Page 18, Lines 329–331: Please consider adding that the lower diversity may also be attributed to the fact that, for more than three decades, hop planting material in Germany has been propagated through certified production of virus-free material in hop nurseries and virus-free hop yards, in accordance with the EPPO standard PM 4/16 (2) – Certification Scheme for Hop.

Thank you for this valuable suggestion. It has been added to the discussion lines 340-345.

Page 19, Line 362: correct to uncertified planting material.

It has been corrected.

Page 19, Line 362-368: ApMV is the most widespread and economically important ilarvirus affecting hop. Numerous studies conducted in the 1990s in the UK, USA, Czech Republic, Poland, and Germany demonstrated its negative impact on hop yield and quality. Several studies have also shown a syn

---

## [Decision Letter · Decision Letter 1]

Dear Dr. Ziebell,

Thank you for submitting your manuscript to PLOS ONE. After careful consideration, we feel that it has merit but does not fully meet PLOS ONE’s publication criteria as it currently stands. Therefore, we invite you to submit a revised version of the manuscript that addresses the points raised during the review process.

I need a rebuttal letter with your submisson

We look forward to receiving your revised manuscript.

Kind regards,

Ahmet Uludag, Ph.D.

Academic Editor

PLOS ONE

Journal Requirements:

Additional Editor Comments:

I think your manuscript is only one step ahead from publishing. Could you please write a good rebuttal for reviewer1's concern and make all rquested changes if possible.

Reviewers' comments:

Reviewer's Responses to Questions

**Comments to the Author**

Reviewer #1: All comments have been addressed

Reviewer #2: All comments have been addressed

2. Is the manuscript technically sound, and do the data support the conclusions?

Reviewer #1: Yes

Reviewer #2: Yes

3. Has the statistical analysis been performed appropriately and rigorously?

Reviewer #1: N/A

Reviewer #2: Yes

4. Have the authors made all data underlying the findings in their manuscript fully available?

Reviewer #1: Yes

Reviewer #2: Yes

5. Is the manuscript presented in an intelligible fashion and written in standard English?

Reviewer #1: Yes

Reviewer #2: Yes

Reviewer #1: On more thing needs to be addressed, in the supplementary table S4, number of reads after QC is very low, which is unexpected. Did you maybe preformed host plant genome subtraction and then you reported the number of reads after that? If that is true please state it in the table and in the manuscript where you describe bioinfo pipeline. If this is not the case, please re-check the data you provided in the supplementary table S4, because it is really suspicious that after QC sometimes you have only about 5% of reads left, this shouldn't be the case if the sequencing went well which based on your other presented data was logical.

Reviewer #2: The manuscript PONE-D-25-26477-R1, entitled "Virus and viroid diversity in hops: Investigating the German hop virome", has been revised according to the reviewers' suggestions. I have no further comments and therefore recommend that the revised manuscript be accepted for publication.

**Do you want your identity to be public for this peer review?** For information about this choice, including consent withdrawal, please see our Privacy Policy

Reviewer #1: No

Reviewer #2: No

---

## [Author Response · Author response to Decision Letter 2]

10 Jul 2025

Reviewer #1:

On more thing needs to be addressed, in the supplementary table S4, number of reads after QC is very low, which is unexpected. Did you maybe preformed host plant genome subtraction and then you reported the number of reads after that? If that is true please state it in the table and in the manuscript where you describe bioinfo pipeline. If this is not the case, please re-check the data you provided in the supplementary table S4, because it is really suspicious that after QC sometimes you have only about 5% of reads left, this shouldn't be the case if the sequencing went well which based on your other presented data was logical.

We also noticed the low number of reads in several datasets after quality control, as presented in S4 Table. We would like to clarify that we did not perform host plant genome subtraction during the data analysis. The low number of retained reads after quality control is can be attributed to two factors:

1- Duplicate reads from library re-sequencing:

As stated in the manuscript, we initially requested that the service provider to generate approximately 10 million paired-end reads per library. According to the service provider, in some cases, fewer reads were obtained initially, so the service re-sequenced the same library to reach the requested number. However, this re-sequencing introduced a high number of technical duplicate reads, which increased the total number of raw reads artificially.

We investigated this issue by analyzing some of our datasets in Geneious Prime with the Dedupe tool vs. 38.84 “Duplicate Read Remover” using the default settings. The results confirmed the presence of a substantial number of duplicate reads, which were subsequently filtered out during normalization with BBNorm alongside other low-quality reads with a Q-score below 20.

2- The implementation of virus/viroid enrichment method:

We performed virus and viroid enrichment (M&M). This method reduces host and microbial RNA that present in the sample. Although this method improves the detection of viruses and viroids, it results in lower total RNA yields. Sequencing providers sometimes misinterpret the low RNA yields as poor quality, so they repeated library sequencing without taking the enrichment strategy into account, and this happened in our case (clarified by the service provider AFTER we obtained the raw reads).

We used only the acquired reads after QC for downstream analysis and found them to be sufficient for detection of partial or full genomes of viruses and viroids. In addition, we were also able to identify other microorganisms and plant sequences. A new column containing the number of “obtained contigs” has been added to S4 Table.

Reviewer #2:

The manuscript PONE-D-25-26477-R1, entitled "Virus and viroid diversity in hops: Investigating the German hop virome", has been revised according to the reviewers' suggestions. I have no further comments and therefore recommend that the revised manuscript be accepted for publication.

Thank you very much for revising our manuscript - we appreciate your valuable feedback.

---

## [Decision Letter · Decision Letter 2]

Virus and viroid diversity in hops, investigating the German hop virome

PONE-D-25-26477R2

Dear Dr. Ziebell,

We’re pleased to inform you that your manuscript has been judged scientifically suitable for publication and will be formally accepted for publication once it meets all outstanding technical requirements.

Kind regards,

Ahmet Uludag, Ph.D.

Academic Editor

PLOS ONE

Additional Editor Comments (optional):

congratulations

Reviewers' comments:

Reviewer's Responses to Questions

**Comments to the Author**

Reviewer #1: All comments have been addressed

Reviewer #2: All comments have been addressed

2. Is the manuscript technically sound, and do the data support the conclusions?

Reviewer #1: Yes

Reviewer #2: Yes

3. Has the statistical analysis been performed appropriately and rigorously?

Reviewer #1: N/A

Reviewer #2: Yes

4. Have the authors made all data underlying the findings in their manuscript fully available?

Reviewer #1: Yes

Reviewer #2: Yes

5. Is the manuscript presented in an intelligible fashion and written in standard English?

Reviewer #1: Yes

Reviewer #2: Yes

Reviewer #1: Authors addressed all comments. I do not have any future questions regarding this manuscript for authors to address.

Reviewer #2: I have no further comments and therefore recommend that the revised manuscript be accepted for publication.

**Do you want your identity to be public for this peer review?** For information about this choice, including consent withdrawal, please see our Privacy Policy

Reviewer #1: No

Reviewer #2: **Yes: ** Sebastjan Radišek, Plant Protection Department, Slovenian Institute of Hop Research and Brewing

---

## [Editor Report · Acceptance letter]

PONE-D-25-26477R2

PLOS ONE

Dear Dr. Ziebell,

I'm pleased to inform you that your manuscript has been deemed suitable for publication in PLOS ONE. Congratulations! Your manuscript is now being handed over to our production team.

Kind regards,

on behalf of

Dr. Ahmet Uludag

Academic Editor

PLOS ONE